# Longitudinal analysis of lipid mediators in post-tuberculosis lung disease identifies significant differences

Peter D. Jackson[1]*, Madeline Helwig[1], Mudarshiru Bbuye[2], Margaret A. Freeberg[1], Thomas H. Thatcher[1], Stella Zawedde[3], Bruce Kirenga[2], William Worodria[2], Krishnarao Maddipati[4], Trishul Siddharthan[5], Joaquin Reyna[6], Patricia J. Sime[1]

1 Department of Pulmonary Critical Care, Virginia Commonwealth University, Richmond, Virginia, United States of America, 2 Makerere Lung Institute, Makerere University, Kampala, Uganda, 3 Makerere Infectious Disease Institute, Makerere University, Kampala, Uganda, 4 Department of Pathology, Wayne State University, Detroit, Michigan, United States of America, 5 Department of Pulmonary & Critical Care, University of Miami, Miami, Florida, United States of America, 6 Department of Bioinformatics, Massey Cancer Center, Virginia Commonwealth University, Richmond, Virginia, United States of America

* jacksonpd@vcu.edu

## Abstract

Post-tuberculosis lung disease (PTLD) is a frequent complication after TB cure, affecting millions globally. Yet, its mechanisms, particularly the role of lipid mediators (LMs) in resolution of inflammation are not clear. This study aimed to identify inflammatory and resolution signatures by comparing longitudinal LM profiles in plasma and exhaled breath condensate (EBC) between TB patients who developed PTLD and those with normal lung recovery.In this prospective cohort study, plasma and EBC were collected at diagnosis and post-TB treatment (generally 6 months). Twenty subjects were selected for analysis following TB treatment, 10 cases (impaired lung function indicative of PTLD) and 10 controls (normal lung function). LM analysis was performed using LC-MS/MS. Group comparisons and longitudinal changes were analyzed using differential abundance analyses (limma) with multiple testing correction with sequential goodness-of-fit adjusted p-values.Baseline plasma analysis showed higher PD1, 12-HEPE, 10-HDoHE, 11-HDoHE, 12-HETE, and 13(14)-EpDPE with significantly lower concentrations of PGA2 in cases vs. controls. Post-treatment, cases had lower AT-RvD6 and 15-oxo-LXA4. Longitudinal plasma analysis with linear modeling showed decreased PD1 in cases during treatment relative to controls. EBC analysis showed no significant differences between group or longitudinal linear comparisons.Distinct systemic lipid mediator profiles and their longitudinal trajectories differ between TB patients who subsequently develop PTLD and those who achieve normal lung recovery. Cases showed an altered LM profile at diagnosis that evolved into a post-treatment state with pro-resolving mediator deficiencies and failed to normalize baseline alterations, suggesting dysregulated resolution pathways in PTLD pathogenesis.

which permits unrestricted use, distribution, and reproduction in any medium, provided the original author and source are credited.

**Data availability statement:** Primary de-identified data used in this manuscript is available online at Peter Jackson (2026). Longitudinal Analysis of Lipid Mediators in Post-Tuberculosis Lung Disease Identifies Significant Differences. BioStudies, S-BSST2862. Retrieved from https://www.ebi.ac.uk/biostudies/studies/S-BSST2862.

**Funding:** This work was supported by the Pulmonary Fibrosis Foundation, CCTR Endowment Fund of the Virginia Commonwealth University to PJ, CTSA (#UL1TR002649) from the National Center for Advancing Translational Sciences to PJ, and NCATS CTSA award #UM1TR004360 to PJ. The NIH Fogarty LAUNCH Development Grant 5D43TW009340-13 to MH. The funders had no role in study design, data collection and analysis, decision to publish, or preparation of the manuscript.

**Competing interests:** I have read the journal's policy and the authors of this manuscript have the following competing interests all research was completed without any input from outside entities. PJ has received consulting fees from Verona Pharmaceuticals. TH has received honoraria from the NIH for grant review. TS has received consulting fees from Apogee Therapeutics, GlaxoSmithKline, Verona Pharmaceuticals, AstraZeneca and speaking honoraria from Sanofi. PS has received consulting fees from Avalyn Pharma, Boehringer Ingelheim, VYNE Therapeutics Inc., Fibrogen and owns stock in Galecto. Additionally, PS has grants and contracts from NIH, UCB pharmaceutical, Roches Genentech, Bristol Myers Squibb, Novomedix and the Ford Foundation.

## Introduction

Tuberculosis (TB), caused by *Mycobacterium tuberculosis* (Mtb) is the leading cause of infectious death globally with over 10 million cases and 1.3 million deaths in 2023 [1]. While cure rates have increased, 40–60% of patients develop chronic respiratory sequelae known as post-tuberculosis lung disease (PTLD) after microbiologic cure [2–5]. PTLD is currently defined as "evidence of chronic respiratory abnormality, with or without symptoms, attributable at least in part to previous pulmonary TB" [3]. The global prevalence of PTLD is alarming, with estimates suggesting that 62–93 million people were living with this condition in 2020 [5,6]. PTLD also reduces quality of life, diminishes economic productivity, and previous TB infection, regardless of PTLD status, leads to approximately 2.9-fold increased mortality [3,7–9].

Despite the profound impact of PTLD, the mechanisms contributing to its development are poorly defined. Pathophysiology is thought to involve persistent inflammation and aberrant tissue repair after the initial inflammatory response to mTB [8]. Recent work has shown that resolution of inflammation is not merely a passive cessation of inflammatory signals but an active, biochemically mediated process to restore tissue homeostasis and function [10]. This process is orchestrated by a complex array of lipid mediators (LMs), including pro-inflammatory prostaglandins and leukotrienes and specialized pro-resolving mediators (SPMs) derived from the polyunsaturated fatty acids eicosapentaenoic acid and docosahexaenoic acid, which include lipoxins, resolvins, protectins and maresins [10–12]. These SPMs are increasingly recognized for their role in counter-regulating pro-inflammatory pathways, promoting debris clearance, and stimulating tissue regeneration [12].

An imbalance between pro-inflammatory LMs and SPMs, or a defect in SPM biosynthesis or signaling, can lead to failed resolution, chronic inflammation, and pathological tissue remodeling, potentially culminating in the manifestations of PTLD [13]. Previous research has noted alterations in LM profiles in TB patients. For instance, the balance between LXA4 (pro-resolving) and prostaglandin $E_2$ (generally pro-inflammatory) influences macrophage responses to Mtb [14,15]. Furthermore, studies have associated dysregulated eicosanoid and SPM levels with TB severity and outcomes in TB meningitis and in TB patients with comorbidities like diabetes [16,17].

Investigating LM profiles in both systemic circulation (plasma) and the local airway environment (non-invasively assessed via exhaled breath condensate, (EBC) [18]) may provide crucial insights into PTLD pathogenesis. While some studies have examined LMs in active TB [14,16], there are no previously published longitudinal data in LM's in patients with PTLD to our knowledge. This work is critical for identifying early LM signatures that may predict PTLD development, provide insights into pathophysiology and potentially identify treatment targets.

This study addresses this knowledge gap by comparing the profiles of eicosanoids and SPMs in plasma and EBC between TB patients who developed PTLD and those with normal lung recovery and identifying differences in LM profile prior to PTLD development and after PTLD is diagnosed.

## Methods

### Study design and enrollment

Within a prospective, longitudinal cohort study at four clinics in Uganda, a sample of 20 adult patients, aged 18 or older, was selected. The study was conducted from May 1, 2022 to December 31, 2023 by a multi-disciplinary team with experience in chronic respiratory disease and TB research. Eligible participants had newly diagnosed pulmonary TB, confirmed with positive sputum acid-fast bacilli (AFB) smear or GeneXpert MTB/RIF assay, and an abnormal chest x-ray (CXR). Exclusion criteria included: TB therapy > 14 days prior to enrollment, self-reported pre-existing chronic lung disease, smoking history > 15 pack-years, active malignancy and pregnancy. While there is likely significant interplay between HIV, TB, and PTLD, this initial pilot study focused on patients without untreated HIV co-infection, facilitating investigation of LM profiles in a less immunologically complex cohort. The study protocol received full ethical approval from the Virginia Commonwealth University IRB HM20022065 and the Makerere Infectious Disease Research Ethics Committee #IDIREC 040/2021. All participants provided written informed consent in Luganda or English before any study-related procedures.

Each participant underwent assessments at two distinct time points. The first assessment, designated as baseline or pre-treatment (Visit 1), occurred at the time of TB diagnosis before or within 14 days of beginning TB therapy. The second assessment (Visit 2 or post-treatment), was conducted after negative sputum testing and completion of TB therapy, generally 6 months from enrollment and included pulmonary function testing (PFT). At both visits, clinical and demographic information, quality of life questionnaires, socioeconomic data, and CXR were collected using standardized questionnaires and medical record review.

PFT's were performed using the EasyOne-Pro Lab platform (ndd, Zurich, Switzerland) by trained staff in accordance with American Thoracic Society/European Respiratory Society (ATS/ERS) guidelines [19]. These tests included spirometry to measure forced expiratory volume in 1 second (FEV1) and forced vital capacity (FVC), lung volumes via nitrogen washout to determine total lung capacity (TLC), and single-breath diffusing capacity for carbon monoxide (DLCO). PFT results were expressed as absolute values, percent predicted, and Z-scores, utilizing the Global Lung Function Initiative (GLI) multi-ethnic reference equations [20].

For lipid mediator analysis, venous blood was collected into EDTA-anticoagulant tubes. Within 30 minutes of collection, these samples were centrifuged at 1300g for 15 minutes. The platelet-poor plasma was then divided into 0.5ml aliquots in cryovials, and immediately stored at -80°C in Uganda until shipping. In parallel, exhaled breath condensate (EBC) was collected as a non-invasive means of sampling the airway lining fluid [18]. This was performed using the RTube™ device following the manufacturer's instructions and ATS/ERS recommendations. Participants breathed tidally through the device for a standardized duration of 15 minutes while wearing a nose clip to prevent nasal air entrainment. The EBC was collected and aliquoted into 0.5ml samples and stored in cryovials at -80°C locally. Shipping on dry ice to VCU was performed by Biocair, a registered biologic sample shipping company, with continuous temperature monitoring and under CDC import permit 20230830-3214A and stored locally at -80°C.

In our study a total of 109 patients completed TB treatment and PFT testing at Visit 2, these subjects were classified as Cases (42 subjects) or Controls (67 subjects). Cases were defined by impaired lung function including one or more of the following: FEV1/FVC ratio Z-score < -1.64; restriction, TLC below 80% of the predicted value; or impaired gas exchange, DLCO below 80% of the predicted value. Controls were defined by the absence of all these PFT abnormalities, thereby demonstrating normal lung function recovery. For this pilot study 10 cases and 10 controls were selected from the larger population to identify LM's of interest and improve cost efficiency. Cases were selected based upon the presence of high quality PFT's at end of treatment (presented here), as well as one-year follow up (samples not yet analyzed due to cost constraints). Additionally, cases without uncontrolled HIV were selected (1 case with HIV had undetectable viral load) and cases without high quality controls were not selected. Cases and controls were sex-matched, controls were age-matched to cases in 10-year blocks to minimize confounding by age. (Fig A in S1 Appendix).

## Lipid mediator analysis

Plasma and EBC samples were thawed on ice under BSL2 + precautions and three volumes of ethanol were added to inactivate any possible Mtb contamination (final ethanol concentration 75%). Plasma is unlikely to contain infectious material but the EBC from enrollment, before antibiotic treatment, could conceivably contain infectious Mtb. After ethanol treatment the samples were refrozen and shipped to Wayne State University for lipidomic analysis. At Wayne State, samples were spiked with a mixture of internal standards for recovery and quantitation. There was no dilution correction performed given lack of gold standard marker for EBC, pilot nature of the study and the risk of bias with correction. The samples were then extracted using solid-phase extraction on C18 columns. Profiling of eicosanoids and SPMs was performed using liquid chromatography-tandem mass spectrometry based on established methodologies [21–24]. The data were collected and MRM transition chromatograms quantitated using Sciex OS 3.3 software. The internal standard signals in each chromatogram were used for normalization for recovery as well as relative quantitation of each analyte.

## Statistical analysis

Baseline demographic characteristics and pulmonary function data were compared between Cases and Controls using independent samples t-tests or Mann-Whitney U tests for continuous variables and are presented in Table 1. To compare LM abundances across conditions and time points, we performed differential abundance analysis using the limma package in R. LM concentrations were log2-transformed prior to analysis to better approximate a normal distribution. For cross-sectional comparisons between treatments, limma was run separately for each time point and sample type (i.e., plasma and EBC). To assess longitudinal changes (post-treatment vs. baseline), limma was used within each treatment group. Lipid concentration data were analyzed using linear mixed-effects models to assess longitudinal changes and experimental group differences in R using the lme4 and lmerTest packages. Low-variance lipid species were removed using the nearZeroVar function (caret package), and each lipid species was analyzed independently. Models included treatment, time, and their interaction as fixed effects, with a random intercept to account for repeated measures. Estimates, standard errors, t-statistics, p-values, and 95% confidence intervals (±1.96 SE) were extracted. From the limma output, we extracted log2 fold changes (logFC) and raw p-values. To adjust for multiple testing, we applied the sequential goodness-of-fit (SGoF) procedure, a method recommended for controlling false discovery in lipidomics [25]. Lipids with SGoF. adj. p value< 0.05 to account for multiple comparison were considered statistically significant, however additional LM's with unadjusted p values < 0.05 are reported. Additional statistical analyses were performed using Microsoft Excel (version 16) and R (version 4.3.3).

## Results

### Study population

For the 20 participants (10 Cases, 10 Controls), baseline characteristics are presented in Table 1. Cases and controls were matched for age and gender and had similar BMI at admission. By definition, cases had either a TLC or DLCO<80% predicted or FEV/FVC z-score<-1.64 using GLI global reference values [26]. Table 1 confirms that indeed the FEV1 and FVC z-score as well as DLCO and TLC percent predicted were significantly lower in cases than controls. FEV1/FVC z-score was lower in cases but not significantly likely due to variation in definitions for cases. The cases contained 2 subjects with diffusion defects (DLCO <80%), 3 with diffusion defects and restriction (DLCO and TLC<80%), 2 with mixed pattern (FEV1/FVC z-score<-1.64 and TLC reduction) and one with diffusion reduction and obstruction (FEV1/FVC z-score<-1.64 and DLCO <80%).

**Table 1. Baseline demographic and clinical characteristics and end of treatment PFT values.**

| Variable: | Cases (n = 10) | Controls (n = 10) | P-value |
|---|---|---|---|
| Age (years), median [IQR] | 30.50 [27.00-35.75] | 30.00 [27.50-37.00] | 0.97 |
| Sex (Male), n (%) | 6/10 (60.0%) | 6/10 (60.0%) | 1 |
| BMI (kg/m²), median [IQR] | 17.00 [15.70-19.02] | 18.80 [18.07-19.57] | 0.161 |
| HIV Positive, n (%) | 1/10 (10.0%) | 0/10 (0.0%) | 1 |
| DLCO (%), median [IQR] | 0.74 [0.57-0.77] | 1.14 [1.10-1.18] | <0.001 |
| TLC (%), median [IQR] | 72.00 [68.00-78.80] | 99.00 [82.25-100.75] | 0.012 |
| FEV1/FVC (z-score), median [IQR] | -1.10 [-1.98--0.15] | -0.12 [-0.60-0.94] | 0.064 |
| FEV1 (z-score), median [IQR] | -2.47 [-3.41--1.87] | 0.14 [-0.42-0.34] | <0.001 |
| FVC (z-score), median [IQR] | -1.83 [-2.94--1.58] | -0.28 [-0.49-0.01] ( | <0.001 |

## Lipid mediator profiles at baseline (TB diagnosis)

**Plasma.** At baseline, significant differences in plasma LM profiles were observed between Cases and Controls (Table 2 and Fig 1). Cases exhibited significantly higher plasma concentrations of protectin D-1 (PD1), 12-HEPE, 10-HDoHE, 11-HDoHE, 12-HETE, and 13(14)-EpDPE with significantly lower concentrations of PGA2 compared to Controls. Additional trends were apparent with numerous Docosanoids (HDoHE below), Eicosanoids (HETE and HETrE) elevated in cases with 15-oxo-LXA4 and AT-RvD6 being insignificantly lower.

**Exhaled breath condensate (EBC).** No statistically significant differences in baseline EBC LM levels were found between Cases and Controls after sequential goodness-of-fit adjustment (all sgof.adj.P.Val > 0.05, Table A in S1 Appendix).

**Table 2. Differences in plasma lipids in cases vs. controls pre-treatment.**

| Lipid | logFC | AveExpr | P.Value | SGoF adj. p | Direction of Change |
|---|---|---|---|---|---|
| PD1 | 0.269 | 0.098 | 0.002 | 0.015 | Higher |
| 12-HEPE | 0.704 | 0.523 | 0.005 | 0.017 | Higher |
| PGA2 | -0.64 | 0.485 | 0.009 | 0.019 | Lower |
| 10-HDoHE | 0.81 | 0.586 | 0.01 | 0.022 | Higher |
| 12-HETE | 1.657 | 2.184 | 0.013 | 0.029 | Higher |
| 13(14)-EpDPE | 0.314 | 0.288 | 0.013 | 0.032 | Higher |
| 11-HDoHE | 0.87 | 0.587 | 0.015 | 0.035 | Higher |
| 13-HDoHE | 0.806 | 0.619 | 0.015 | 0.052 | Insignificantly Higher |
| 15-oxo LXA4 | -0.592 | 0.807 | 0.017 | 0.062 | Insignificantly Lower |
| 8-HDoHE | 0.691 | 0.515 | 0.019 | 0.064 | Insignificantly Higher |
| 14-HDoHE | 0.849 | 0.637 | 0.019 | 0.069 | Insignificantly Higher |
| 8-HETrE | 0.703 | 0.642 | 0.022 | 0.075 | Insignificantly Higher |
| Maresin2 | 0.282 | 0.142 | 0.029 | 0.256 | Insignificantly Higher |
| 8-HETE | 0.958 | 0.907 | 0.029 | 0.26 | Insignificantly Higher |
| AT-RvD6 | -0.506 | 0.627 | 0.032 | 0.363 | Insignificantly Lower |
| AT-PD1 | 0.167 | 0.087 | 0.035 | 0.366 | Insignificantly Higher |
| iPF-VI | -0.503 | 0.711 | 0.046 | 0.438 | Insignificantly Lower |

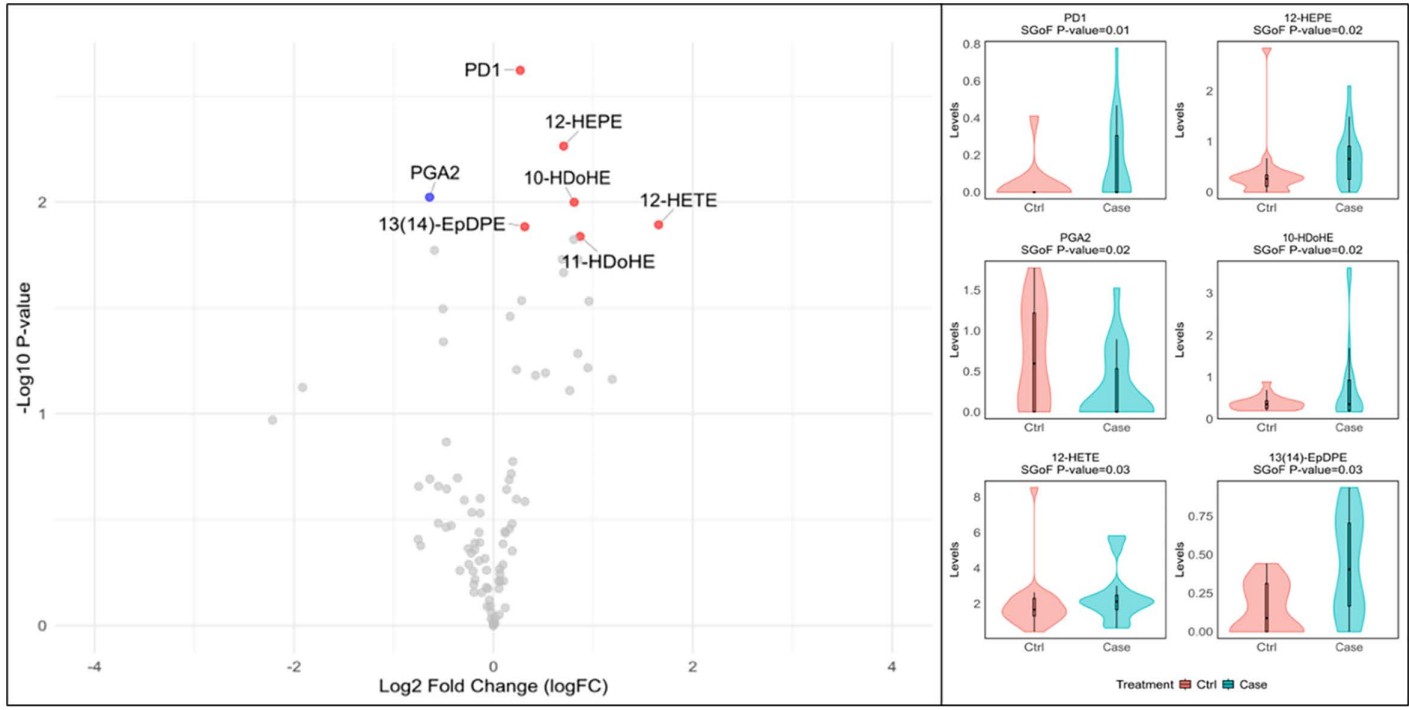

PLOS Global Public Health logo

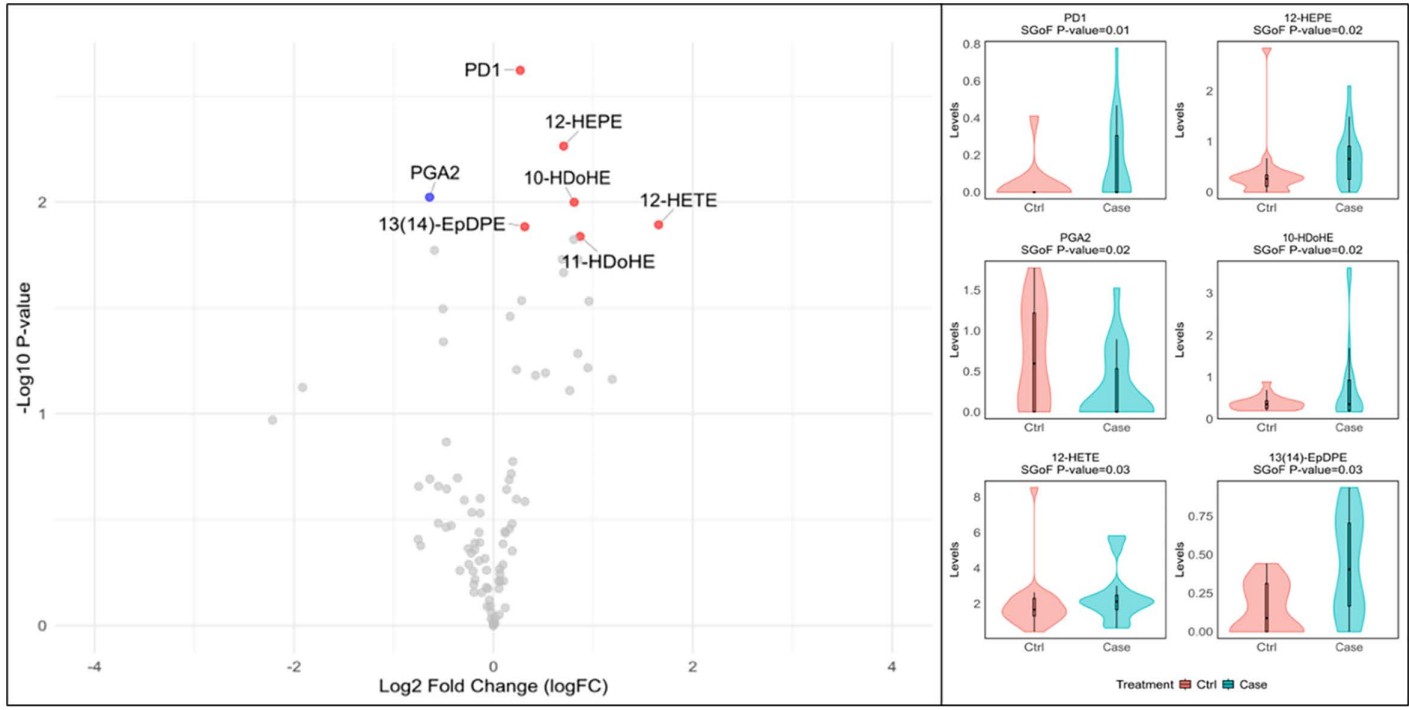

**Fig 1. Volcano and violin plot showing significant differences in plasma LMs in cases vs. controls pre-treatment.** Within the volcano plot red dots indicate significantly elevated LM species and blue indicate significantly lower LM species within cases based on SGoF. adj. p value<0.05. Grey dots within the volcano plot represent LM species that are not significantly different between cases and controls. The violin plots represent the six LM species with the most significant differences amongst cases and controls, the y axis represents concentration levels in ng/mL and scales are adjusted for each species to improve visualization.

### Lipid mediator profiles post-treatment (~6 months)

**Plasma.** During the post-treatment visit there were additional significant differences in plasma LM profiles between the controls and PTLD subjects (sgof.adj.P.Val<0.05) (Fig 2 and Table 3). Cases had significantly lower plasma concentrations of AT-RvD6 and 15-oxo-LXA4 compared to Controls. There were trends which were not significant after adjustment that are noted in Table 3 including lower 15-OxoETE and LXB4.

**Table 3. Plasma LM's in cases vs. controls after TB treatment.**

| Lipid | logFC | AveExpr | P.Value | SGoF adj. p | Direction of Change |
|---|---|---|---|---|---|
| AT-RVD6 | -1.472 | 0.627 | >0.001 | 0.039 | Lower |
| 15-oxo LXA4 | -1.13 | 0.807 | >0.001 | 0.042 | Lower |
| 15-OxoETE | -0.583 | 0.773 | 0.006 | 0.133 | Insignificantly Lower |
| 13-OxoODE | -2.964 | 2.423 | 0.007 | 0.134 | Insignificantly Lower |
| 9-HEPE | 0.311 | 0.149 | 0.01 | 0.238 | Insignificantly Higher |
| 8(9)-EpETrE | 0.309 | 0.409 | 0.025 | 0.626 | Insignificantly Higher |
| 13(14)-EpDPE | 0.26 | 0.288 | 0.037 | 0.706 | Insignificantly Higher |
| LXB4 | -1.078 | 1.368 | 0.038 | 0.737 | Insignificantly Lower |

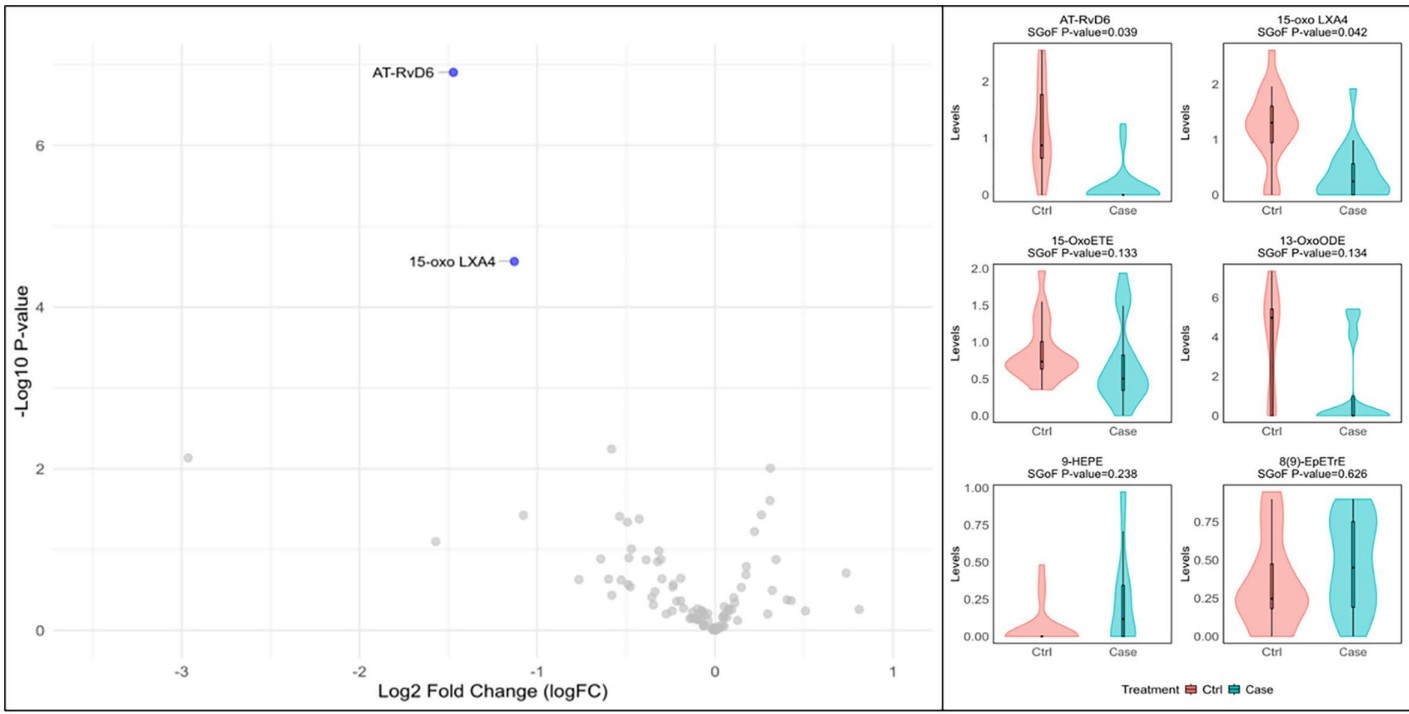

**Fig 2. Volcano and violin plot showing significant differences in plasma LMs in cases vs. controls post TB treatment.** Within the volcano plot red dots indicate significantly elevated LM species and blue indicate significantly lower LM species within cases based on SGoF. adj. p < 0.05. Grey dots within the volcano plot represent LM species that are not significantly different between cases and controls. The violin plots represent the six LM species with the most significant differences amongst cases and controls, the y axis represents concentration levels in ng/mL and scales are adjusted for each species to improve visualization.

**Exhaled breath condensate (EBC).** Similar to baseline, no statistically significant differences in post-treatment EBC LM levels were observed between Cases and Controls after sequential goodness-of-fit adjustment (all sgof. adj.P.Val > 0.117) (Table B in S1 Appendix).

## Longitudinal changes in lipid mediators by linear modeling with interaction for time and group in cases vs. controls

**Plasma.** To evaluate how treatment influenced lipid mediator (LM) profiles differently between cases and controls, we analyzed the interaction between group (Case vs. Control) and time (Baseline to Post-Treatment). One LM, PD1 demonstrated a significant decrease in cases during treatment relative to controls (SGoF. adj. p < 0.001). There were notable trends for AT-RvD6

15-OxoETE and 12-OxoETE having less increase relative to controls but these were not significant after sequential goodness of fit adjustment (Fig 3 and Table 4).

**Exhaled breath condensate (EBC).** No statistically significant longitudinal changes using linear mixed effects model in cases vs. controls were detected for any measured EBC LM after sequential goodness-of-fit adjustment (all sgof. adj.P.Val > 0.05), (Table C in S1 Appendix).

**Table 4. Linear mixed-effects model of longitudinal changes and differences in plasma by group.**

| Lipid | Interaction Estimate | Interaction p-value | Interaction SGoF. adj. p | Direction of Change |
|---|---|---|---|---|
| PD1 | -0.255 | <0.001 | 0.001 | Lower |
| AT-RvD6 | -1.738 | 0.005 | 0.061 | Insignificantly Lower |
| 15-OxoETE | -1.199 | 0.007 | 0.071 | Insignificantly Lower |
| 12-OxoETE | -0.15 | 0.016 | 0.071 | Insignificantly Lower |
| 8(9)-EpETrE | 0.363 | 0.019 | 0.08 | Higher |
| 11-12-DiHETrE | 0.375 | 0.02 | 0.097 | Higher |
| Maresin2 | -0.295 | 0.033 | 0.097 | Insignificantly Lower |
| PD1(n-3-DPA) | -0.191 | 0.046 | 0.101 | Insignificantly Lower |
| Tetranor-12-HETE | -0.2 | 0.048 | 0.103 | Insignificantly Lower |

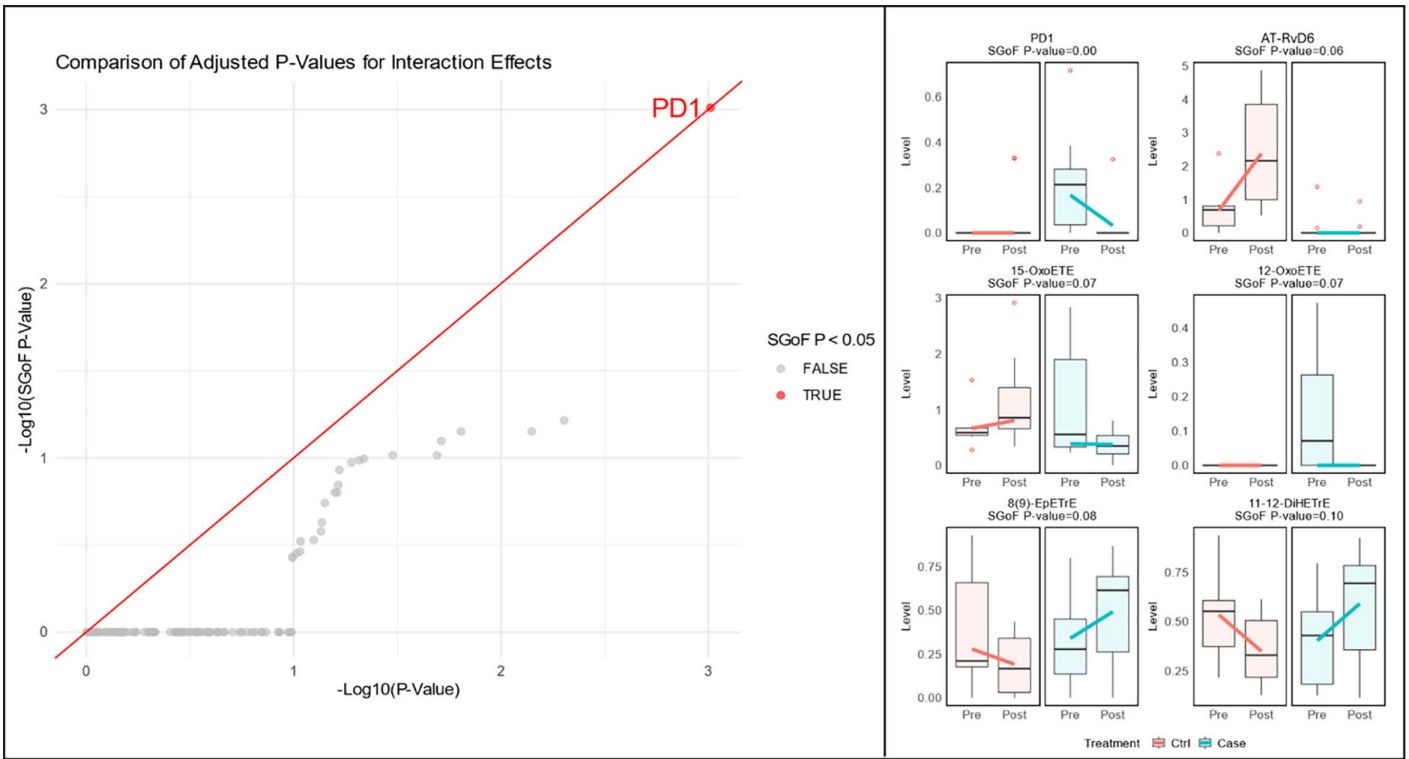

**Fig 3. Comparison of sequential goodness of fit p values for interaction effects between baseline and 6-month time point.** Highlighted point in red signifies significantly different trajectory of lipid levels during treatment. Points in gray are non-significant lipids. The 6 bar plots on the right represent lipid levels on the y-axis in ng/ml in cases and controls with interaction for time and individual patient. The 6 bar plots correspond with the lowest SGoF. adj. p values with PD1 being significantly different in cases and controls.

## Discussion

This study provides potential evidence for distinct systemic LM profiles and longitudinal trajectories associated with PTLD development in patients treated for pulmonary TB. Utilizing sequential goodness-of-fit adjustment for multiple comparisons, we identified significant differences in plasma LM signatures between individuals who developed PTLD and those with normal lung recovery, both at the time of TB diagnosis, prior to identifying PTLD and post-treatment when PTLD

was diagnosed. Furthermore, significant longitudinal changes in specific plasma LMs using delta-delta comparison were observed in cases vs. controls. While there were interesting trends, EBC analysis did not reveal significant differences after adjustment in the local environment.

The baseline plasma profile in patients who developed PTLD was characterized by higher levels of PD1, 12-HEPE, 10-HDoHE, 11-HDoHE, 12-HETE, and 13(14)-EpDPE, alongside lower $PGA_2$, compared to patients with healthy recovery. This early signature suggests a potential dysregulation of the inflammatory/resolution balance in cases at TB diagnosis [10,27,28]. The elevation of PD1, a pro-resolving SPM, is initially counterintuitive in cases. However, PD1 is reported to inhibit production of TNFα and IFNγ [13,29], cytokines whose finely tuned balance is critical in the initial response to mTB invasion. Elevated PD1 too early in the disease may lead to inadequate mycobacterial clearance, prolonged inflammation and poorer prognosis [30,31]. This hypothesis is supported by data showing that PD1 and downstream metabolites of HDoHE's (resolvins and maresins) activate the ALX/FPR2 receptor on macrophages leading to an M2 phenotype. While M2 macrophages are typically pro-resolving, they also promote persistence of mTB infection, which could result in chronic inflammation and PTLD [32,33].

The presence of 12-HEPE in cases at the baseline visit (prior to development of PTLD) may have similar effects. 12-HEPE has been shown to reduce neutrophil infiltration in skin in allergy potentially also causing poor neutrophil clearance of mTB early in disease [34]. 12-HETE is pro-inflammatory and has been shown to worsen outcomes in infection and leads to bronchial injury in asthma patients [35]. Elevated levels of these mediators at baseline may pre-dispose these patients to develop PTLD, however, given the competing pro and anti-inflammatory signals larger studies and lab based work to untangle these findings are needed.

Post-treatment plasma profiles also highlight significant LM differences within PLTD. Notably, PTLD subjects exhibited significantly lower concentrations of key SPMs such as AT-RvD6, an aspirin-triggered resolvin D6 pathway marker, and 15-oxo-LXA4, a metabolite indicating lipoxin activity. This pattern implicates an impaired capacity for active inflammation resolution and a failure to restore biochemical homeostasis in the pathophysiology of PTLD [10,15,28]. Deficiencies in SPMs like resolvins and lipoxins are known to contribute to persistent inflammation and impaired tissue repair in other chronic inflammatory diseases and LXA4 has been shown to be involved in macrophage death, clearance of apoptotic cells and poor control of Mtb [15,36,37].

Longitudinal analysis using interaction effects provides a crucial temporal dimension to these findings, identifying how systemic LM trajectories diverge during TB recovery in PTLD and controls. Within the case group, PD1 was higher at baseline and decreased during treatment in cases compared to a relatively stable low expression in controls. As previously discussed, this finding may suggest a role for PD1 in PTLD development through inhibiting inflammatory response and shifting macrophages to an M2 phenotype early in the disease; PD1's decrease during treatment is not surprising given decrease in inflammation, however, in cases it is possible that this decrease is also maladaptive with PD1 potentially having a role for mitigation of chronic inflammation within cases. Other notable LM trends during treatment included AT-RvD6, 15-OxoETE and 12-OxoETE having less increase or relative decrease compared to controls. This decrease in key regulatory lipids in cases compared to increases in controls further supports the concept of poorly timed resolution leading to chronic inflammation while controls have increasing levels of these pro-resolving LM's throughout treatment and more effective resolution. This failure to actively switch towards, and sustain, a pro-resolving phenotype, characterized by SPM expression, may suggest persistent inflammation and aberrant repair leading to PTLD. This aligns with previous findings in other chronic inflammatory conditions where SPM deficiency correlates with disease persistence and severity [29].

The consistent lack of significant findings in EBC, both cross-sectionally and longitudinally after appropriate statistical adjustment is not necessarily surprising in this small sample. EBC was collected on the assumption that direct sampling from the lung environment could provide insights into mechanisms of disease within the local environment. the lack of significant lipid mediator (LM) differences in EBC after statistical adjustment must be interpreted within the context of matrix dilution. Quantitative comparison revealed a profound 'dilution effect'; across the 147 lipid species analyzed, 84.4% of

lipids in the EBC matrix had median concentrations at the absolute lower limit of detection (1 x 10$^{-5}$ ng/mL), compared to only 49.7% in the plasma. Despite this dilution, the technical validity of our EBC findings is supported by the high stability of our internal standards, which exhibited mean quality scores exceeding 0.92 and a low Coefficient of Variation (<15%) across the EBC cohort. These data suggest that the null findings in EBC are not the result of technical failure or poor recovery, but rather reflect a lack of detectable variance in the highly dilute local airway environment. It should be mentioned that while EBC dilution is present, it is also possible that the kinetics of LM changes in airway lining fluid differ from those in systemic circulation, or that EBC reflects primarily epithelial cells but not other lung compartments (e.g., parenchyma, interstitium) that may be more relevant to PTLD. The significant plasma findings suggest that PTLD is accompanied by systemic LM dysregulation, a concept supported by evidence of extra-pulmonary TB sequela including cardiac and neurologic complications [3]. Our group's previous work has shown differences in EBC SPMs in other lung diseases like COPD [36,38], suggesting ongoing potential, however, larger sample sizes or different approaches (e.g., bronchoscopy) may be necessary in PTLD.

Strengths of this study include its prospective longitudinal design, objective classification of PTLD based on comprehensive PFTs using GLI criteria, and parallel analysis of LMs in both plasma and EBC. The inclusion of detailed longitudinal analysis and the application of appropriate statistical correction for multiple comparisons are also strengths.

Limitations include the small sample size (n = 20), which limits statistical power, particularly for the EBC analysis where mediator concentrations are low making detection of small but biologically relevant differences challenging. The low concentrations of LMs in EBC, coupled with inter-individual variability, likely contributed to the lack of significant findings after correction. We were not able to adjust for all potential confounders, although age and gender matching was performed. The small sample did not make it feasible to examine LM associations with specific PFT patterns, comorbidities, or other variables of interest.

In conclusion, our findings support a role for a dysregulated systemic inflammatory response to Mtb infection at TB diagnosis and alterations in LM trajectories during TB treatment in PTLD. This difference in initial and longitudinal SPM concentration may lead to chronic inflammation, ineffective tissue repair, and the establishment of PTLD. This supports future research in larger cohorts to validate PD1, AT-RvD6 and other key LM's association with PTLD and offers potential avenues for predictive biomarkers, translational studies of PTLD pathogenesis and the development of novel therapeutic strategies to promote lung health in TB survivors.

## Supporting information

**S1 Appendix.** Fig A: CONSORT diagram detailing overall patient recruitment and selection of cases and controls for lipidomic analysis. Table A: Pre-Treatment Cases vs. Controls EBC Lipid Metabolites. Table B: Post-Treatment Cases vs. Post-Treatment Controls EBC Lipid Metabolites. Table C: Linear mixed-effects model to assess longitudinal changes and differences in EBC. Interaction estimates and p values for the ten most significant LM's by adjusted sequential goodness of fit p-value. S1 Text A: Lipid Glossary.
(DOCX)

## Author contributions

**Conceptualization:** Peter Durham Jackson, Bruce Kirenga, Trishul Siddharthan, Patricia J Sime.

**Data curation:** Peter Durham Jackson, Madeline Helwig, Margaret A Freeberg, Krishnarao Maddipati, Joaquin Reyna.

**Formal analysis:** Peter Durham Jackson, Madeline Helwig, Margaret A Freeberg, Joaquin Reyna.

**Funding acquisition:** Peter Durham Jackson.

**Investigation:** Peter Durham Jackson, Madeline Helwig, Bbuye Mudarshiru, Stella Zawedde, Krishnarao Maddipati.

**Methodology:** Peter Durham Jackson, Madeline Helwig, Bbuye Mudarshiru, Margaret A Freeberg, Thomas H Thatcher, Stella Zawedde, Bruce Kirenga, Krishnarao Maddipati, Trishul Siddharthan, Joaquin Reyna.

**Project administration:** Peter Durham Jackson, Madeline Helwig, Bbuye Mudarshiru, William Worodria.

**Resources:** Thomas H Thatcher, Trishul Siddharthan, Patricia J Sime.

**Software:** Joaquin Reyna.

**Supervision:** Peter Durham Jackson, Bbuye Mudarshiru, Thomas H Thatcher, Stella Zawedde, Bruce Kirenga, William Worodria, Krishnarao Maddipati, Trishul Siddharthan, Patricia J Sime.

**Validation:** Peter Durham Jackson.

**Visualization:** Peter Durham Jackson, Joaquin Reyna.

**Writing – original draft:** Peter Durham Jackson.

**Writing – review & editing:** Peter Durham Jackson, Madeline Helwig, Bbuye Mudarshiru, Margaret A Freeberg, Thomas H Thatcher, Stella Zawedde, Bruce Kirenga, William Worodria, Krishnarao Maddipati, Trishul Siddharthan, Joaquin Reyna, Patricia J Sime.

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
