## [Decision Letter · Decision Letter 0]

28 Dec 2025

PGPH-D-25-03241

Longitudinal Analysis of Lipid Mediators in Post-Tuberculosis Lung Disease Identifies Significant Differences.

Dear Dr. Jackson,

Thank you for submitting your manuscript to PLOS Global Public Health. After careful consideration, we feel that it has merit but does not fully meet PLOS Global Public Health’s publication criteria as it currently stands. Therefore, we invite you to submit a revised version of the manuscript that addresses the points raised during the review process.

We look forward to receiving your revised manuscript.

Kind regards,

Paulo Jorge Gonçalves de Bettencourt, Ph.D.

Academic Editor

Journal Requirements:

1. Please provide a detailed online Financial Disclosure statement. This is published with the article. It must therefore be completed in full sentences and contain the exact wording you wish to be published.

a) Please clarify all sources of financial support for your study. List the grants, grant numbers, and organizations that funded your study, including funding received from your institution. Please note that suppliers of material support, including research materials, should be recognized in the Acknowledgements section rather than in the Financial Disclosure.

b) State the initials, alongside each funding source, of each author to receive each grant. For example: “This work was supported by the National Institutes of Health (####### to AM; ###### to CJ) and the National Science Foundation (###### to AM).”

c) State what role the funders took in the study. If the funders had no role in your study, please state: “The funders had no role in study design, data collection and analysis, decision to publish, or preparation of the manuscript.”

For more information, please go to our submission guidelines:

https://journals.plos.org/globalpublichealth/s/submission-guidelines#loc-financial-disclosure-statement

2. Please ensure that the funders and grant numbers match between the Financial Disclosure field and the Funding Information tab in your submission form. Note that the funders must be provided in the same order in both places as well.

3. Please send a completed ‘Competing Interests’ statement, including any COIs declared by your co-authors. Please declare all competing interests beginning with the statement “I have read the journal's policy and the authors of this manuscript have the following competing interests:”

For more information, please go to our submission guidelines:

https://journals.plos.org/globalpublichealth/s/submission-guidelines#loc-competing-interests

4. In the online submission form, you indicated that “All data is available by request. Ongoing work to expand on these findings is underway and data will not be uploaded as open access until all samples have been analyzed with additional analysis and correlation with other patient level factors.”.

a) In a public repository,

b) Within the manuscript itself, or

d) Uploaded as supplementary information.

For further assistance, you may go to: http://journals.plos.org/globalpublichealth/s/data-availability

5. Please ensure that the Author List in your manuscript file matches the Author List in the online submission form, including initials and order.

Please use either initials or full names in both the manuscript and online submission form.

6. Please provide separate main figure files in .tif or .eps format only and remove any figures embedded in your manuscript file. Please also ensure that all files are under our size limit of 10MB. Please leave the figure captions in the manuscript.

7. We notice that your supplementary tables are included in the manuscript file. Please remove them and upload them with the file type ‘Supporting Information’. Please ensure that each Supporting Information file has a legend listed in the manuscript before or after the references list.

8. We do not publish any copyright or trademark symbols that usually accompany proprietary names, eg (R), (C), or TM (e.g. next to drug or reagent names). Please remove all instances of trademark/copyright symbols throughout the text, including ™ on page 4.

Additional Editor Comments (if provided):

Reviewers' comments:

Reviewer's Responses to Questions

**Comments to the Author**

1. Does this manuscript meet PLOS Global Public Health’s publication criteria? Is the manuscript technically sound, and do the data support the conclusions? The manuscript must describe methodologically and ethically rigorous research with conclusions that are appropriately drawn based on the data presented.? Is the manuscript technically sound, and do the data support the conclusions? The manuscript must describe methodologically and ethically rigorous research with conclusions that are appropriately drawn based on the data presented.

Reviewer #1: Yes

Reviewer #2: Yes

2. Has the statistical analysis been performed appropriately and rigorously?

Reviewer #1: Yes

Reviewer #2: I don't know

3. Have the authors made all data underlying the findings in their manuscript fully available (please refer to the Data Availability Statement at the start of the manuscript PDF file)?

The PLOS Data policy requires authors to make all data underlying the findings described in their manuscript fully available without restriction, with rare exception. The data should be provided as part of the manuscript or its supporting information, or deposited to a public repository. For example, in addition to summary statistics, the data points behind means, medians and variance measures should be available. If there are restrictions on publicly sharing data—e.g. participant privacy or use of data from a third party—those must be specified.requires authors to make all data underlying the findings described in their manuscript fully available without restriction, with rare exception. The data should be provided as part of the manuscript or its supporting information, or deposited to a public repository. For example, in addition to summary statistics, the data points behind means, medians and variance measures should be available. If there are restrictions on publicly sharing data—e.g. participant privacy or use of data from a third party—those must be specified.

Reviewer #1: Yes

Reviewer #2: Yes

4. Is the manuscript presented in an intelligible fashion and written in standard English?

Reviewer #1: Yes

Reviewer #2: Yes

Reviewer #1: Jackson and colleagues present a really interesting report on possible mediators of post tuberculosis lung disease. Although the prospective clinical study has a relatively low number of patients, it provides clues for follow-up mechanistic studies on the role of lipid mediators in PTLD.

I think the paper would benefit from the clarification of the following points:

1. Am I correct in assuming that the pulmonary function parameters in Table 1 refer to values collected at visit 2 (post-treatment)? Since the authors refer to it as baseline values, my first impression was that they refer to pre-treatment levels. This should me made clearer for a more general readership.

2. What was the difference in pulmonary function parameters from visit 1 to visit 2 in cases and controls?

3. Is there any information on the lesion extent/type pre-treatment between cases and controls? Could disease severity be linked to the observed patterns?

4. Do the authors think that the set of initial conditions (pre-treatment) is the major determinant of PTLD outcomes, or the immune events that occur during treatment are more important? I ask this because, for example, AT-RvD6 is similar pre-treatment but then becomes one of the major differences. Is AT-RvD6 part of the cause or a consequence?

Reviewer #2: Novel pilot LM data in PTLD with rigorous LC-MS/MS and elegant PD1 paradox hypothesis. MAJOR issues: CONSORT flow for 10920 selection bias, PTLD definition mixing phenotypes, delta-delta stats justification, and tempering causal language (suggests vs demonstrates). MINOR: Typos ('in to'), reposition EBC as exploratory, etc.

**Do you want your identity to be public for this peer review?** For information about this choice, including consent withdrawal, please see our Privacy Policy..

Reviewer #1: **Yes:**Tiago BeitesTiago BeitesTiago BeitesTiago Beites

Reviewer #2: **Yes:**Boadu AABoadu AABoadu AABoadu AA

---

## [Editor Report · Decision Letter 1]

23 Feb 2026

Longitudinal Analysis of Lipid Mediators in Post-Tuberculosis Lung Disease Identifies Significant Differences.

PGPH-D-25-03241R1

Dear Dr Jackson,

We are pleased to inform you that your manuscript 'Longitudinal Analysis of Lipid Mediators in Post-Tuberculosis Lung Disease Identifies Significant Differences.' has been provisionally accepted for publication in PLOS Global Public Health.

Best regards,

Paulo Jorge Gonçalves de Bettencourt, Ph.D.

Academic Editor